# Automated brain tumor diagnostics: Empowering neuro-oncology with deep learning-based MRI image analysis

Subathra Gunasekaran[1], Prabin Selvestar Mercy Bai[2], Sandeep Kumar Mathivanan[3], Hariharan Rajadurai[4], Basu Dev Shivahare[3], Mohd Asif Shah 📵 [5,6,7] *

1 Department of Computer Science and Engineering, Sathyabama Institute of Science and Technology, Chennai, India, 2 School of Computer Science and Engineering, Vellore Institute of Technology, Vellore, Tamil Nadu, India, 3 School of Computer Science and Engineering, Galgotias University, Greater Noida, India, 4 School of Computing Science and Engineering, VIT Bhopal University, Sehore, India, 5 Faculty of Kebri Dehar University, Somali, Ethiopia, 6 Division of Research and Development, Lovely Professional University, Phagwara, Punjab, India, 7 Centre of Research Impact and Outcome, Chitkara University Institute of Engineering and Technology, Chitkara University, Rajpura, Punjab, India

* ohaasif@gmail.com

**Data Availability Statement:** Used publicly available database, and no human data/sample

## Abstract

Brain tumors, characterized by the uncontrolled growth of abnormal cells, pose a significant threat to human health. Early detection is crucial for successful treatment and improved patient outcomes. Magnetic Resonance Imaging (MRI) is the primary diagnostic tool for brain tumors, providing detailed visualizations of the brain's intricate structures. However, the complexity and variability of tumor shapes and locations often challenge physicians in achieving accurate tumor segmentation on MRI images. Precise tumor segmentation is essential for effective treatment planning and prognosis. To address this challenge, we propose a novel hybrid deep learning technique, Convolutional Neural Network and ResNeXt101 (ConvNet-ResNeXt101), for automated tumor segmentation and classification. Our approach commences with data acquisition from the BRATS 2020 dataset, a benchmark collection of MRI images with corresponding tumor segmentations. Next, we employ batch normalization to smooth and enhance the collected data, followed by feature extraction using the AlexNet model. This involves extracting features based on tumor shape, position, shape, and surface characteristics. To select the most informative features for effective segmentation, we utilize an advanced meta-heuristics algorithm called Advanced Whale Optimization (AWO). AWO mimics the hunting behavior of humpback whales to iteratively search for the optimal feature subset. With the selected features, we perform image segmentation using the ConvNet-ResNeXt101 model. This deep learning architecture combines the strengths of ConvNet and ResNeXt101, a type of ConvNet with aggregated residual connections. Finally, we apply the same ConvNet-ResNeXt101 model for tumor classification, categorizing the segmented tumor into distinct types. Our experiments demonstrate the superior performance of our proposed ConvNet-ResNeXt101 model compared to existing approaches, achieving an accuracy of 99.27% for the tumor core class with a minimum learning elapsed time of 0.53 s.

used in the study: https://www.kaggle.com/datasets/awsaf49/brats2020-training-data.

**Funding:** The author(s) received no specific funding for this work.

**Competing interests:** NO authors have competing interests.

## 1. Introduction

The brain constitutes a vital organ comprised of neural cells, supported by tissues like glial cells and meninges. Harm to these specific brain components is irreversible and can result in serious conditions, including life-threatening brain tumors. Anomalously developing cells within the human brain give rise to these tumors [1]. The current prevalence of malignant tumors is alarming, profoundly affecting individuals and the community. Magnetic Resonance Imaging (MRI) stands as the foremost clinical imaging method for detecting brain cancers. MRI proves to be a secure and non-invasive diagnostic technology, providing more comprehensive data on brain tissues than computed tomography scans [2]. The precision in accurately delineating brain tumors from clinical images is not merely a technical necessity but forms the crux of establishing a robust and interpretable foundation for medical diagnosis and subsequent therapeutic interventions [3]. The meticulous segregation of brain tumors stands as a pivotal phase in both the diagnostic and therapeutic realms of brain-related disorders. However, achieving this level of precision remains an intricate challenge, given the myriad factors contributing to the complexity of the task [4]. Variations in tumor morphology, size, and location, coupled with indistinct borders, pose formidable obstacles in the pursuit of precise segmentation. In the evolutionary landscape of medical imaging, several methodologies have emerged over the past few decades, each vying to enhance the accuracy and efficiency of brain tumor segmentation [5]. Traditional techniques, reliant on manually crafted features and the application of Machine Learning (ML) models such as Support Vector Machines (SVMs) and Random Forests (RFs), often fall short in achieving optimal performance. In stark contrast, the advent of Deep Learning (DL) techniques in medical image segmentation has elicited substantial interest [6]. The remarkable outcomes attained through DL, particularly in the detection and characterization of target structures within images, have propelled it to the forefront of innovative approaches [7]. The adoption of segmentation methods in medical applications is not solely contingent on their technical prowess but is intricately entwined with the level of user engagement and the ease of implementation. In this context, manual segmentation of brain tumors, while offering a meticulous approach, proves to be a time-intensive endeavour [8]. Researchers are compelled to physically demarcate the Region of Interest (ROI) on MRI segments, leveraging sophisticated graphical user interface tools. However, this manual undertaking, despite its precision, is not immune to the pitfalls of human error, encompassing both inter and intra-variations [9]. As the pursuit of accurate brain tumor segmentation continues, striking a balance between technological advancements and user-friendly applications becomes paramount in ensuring widespread and effective adoption in the medical domain. Implementing an autonomous strategy for the segmentation of brain tumors presents a promising avenue to alleviate the challenges associated with human errors, offering resilience against external influences like disturbances and the mental state of healthcare practitioners [10]. As researchers delve into the realm of pertinent studies, a plethora of recent investigations have successfully unveiled a myriad of efficient automated systems. In light of these advancements, it becomes evident that the responsibilities of a professional tasked with the development of an autonomous segmentation system extend beyond the sole emphasis on constructing a model proficient in autonomous learning and segmentation [11]. While the autonomy of the model is crucial, it is equally imperative to channel efforts into broader enhancements. This involves a strategic focus on refining the architecture of the model, not only to bolster its autonomous capabilities but also to optimize resource utilization [12]. The quest for innovation should include measures to streamline the system's efficiency, making judicious use of available resources and minimizing unnecessary computational burdens. Furthermore, the holistic development of an independent segmentation system should prioritize

user-centric improvements. Enhancing accessibility for users becomes a paramount consideration, ensuring that the system is not only technologically sophisticated but also user-friendly [13]. This entails refining the graphical user interface, simplifying operational procedures, and creating an environment that fosters seamless interaction between the autonomous model and its users. In essence, the evolution of autonomous segmentation strategies transcends the confines of mere technical prowess. It encompasses a multidimensional approach that amalgamates autonomous learning with architectural refinement, resource optimization, and user accessibility enhancements. By navigating this comprehensive pathway, professionals can cultivate segmentation systems that not only exhibit autonomy and accuracy but also cater to the practical needs and usability preferences of the healthcare community [14].

The contribution of research methodology introduced in this investigation presents a novel approach by integrating advanced deep learning techniques, specifically Convolutional Neural Networks (CNNs) enhanced with ResNeXt101 architecture, to address the challenges of precise segmentation and proficient classification in brain tumor analysis. Key contributions of this study include:

- Proposing a sophisticated architecture that combines deep CNN principles with ResNeXt101, aiming to effectively segment and classify brain tumors in MRI images.

- Leveraging MRI datasets from the widely recognized BraTs 2020 datasets, which serve as standard benchmarks in the field of brain tumor research.

- Employing batch normalization techniques during the preprocessing phase to ensure the optimal preparation of MRI data, enhancing the model's ability to learn and generalize.

- Utilizing the AlexNet model for feature extraction from meticulously pre-processed MRI images, enabling the identification of salient features crucial for tumor segmentation and classification.

- Employing the Advanced Whale Optimization (AWO) methodology for feature selection, strategically selecting optimal features that bolster the effectiveness of segmentation and classification efforts.

Overall, this research methodology introduces a comprehensive framework that integrates cutting-edge deep learning techniques with established methodologies, aiming to advance the state-of-the-art in brain tumor analysis. The structure of the article is as follows: Section II presents an overview of existing methods for brain tumor detection and classification. Section III delves into the details of the proposed methodology. Section IV showcases the simulation results, demonstrating the effectiveness of the proposed approach. Finally, Section V concludes the paper, summarizing the key findings and outlining future directions.

## 2. Related work

The author introduced a novel hybrid model, combining Convolutional Neural Network (CNN) and Support Vector Machine (SVM), for both the identification and categorization of brain tumors. This study employs the publicly accessible BraTs 2015 dataset. Subsequently, a threshold-based segmentation algorithm is applied for segmentation purposes. To classify brain MRI images, the labelled segmented features are input into the hybrid CNN and SVM methods. The proposed amalgamation of CNN and SVM demonstrates its efficacy, yielding superior models with a classification accuracy of 98.4%. However, upon considering additional evaluation criteria such as Positive Predictive Value (PPV) and False Positive Value (FPV), the classification performance across CNN, SVM, and the hybrid CNN-SVM remains comparable

[15]. The researcher established a groundbreaking Hybrid Weights Alignment with Multi-Dilated Attention Network (Hybrid-DANet) to facilitate automated segmentation. The efficacy of the Hybrid-DANet is scrutinized using two renowned datasets, namely BraTS 2017 and 2018. Subsequently, an augmentation is introduced in the form of a Multi-Channel Multi-Scale (MCS) component, complementing the fundamental structure. On the BraTS 2018 dataset, Hybrid Weights Alignment with Multi-Dilated Attention Network (HWADA) attains commendable Dice Similarity Coefficient (DSC) results, specifically 0.892, 0.764, and 0.680 for the Whole Tumor (WT), Tumor Core (TC), and Enhanced Tumor (ET), respectively. This achievement is attributed to the extraction of features ranging from basic to deeper characteristics. It's worth noting that, in this study, the evaluation of the proposed model is limited to the encoder component [16]. The proposed technique demonstrated a notable enhancement in brain image classification when applied to a predefined input dataset. The research utilized MRI scans from the REMBRANDT dataset, comprising 620 testing and 2480 training sets. Results indicate that the newly introduced method outperformed its predecessors. In a comparative analysis, the proposed CRNN strategy was pitted against BP, U-Net, and ResNet—three widely adopted classification approaches in current practice. Notably, for brain tumor classification, the proposed system exhibited outcomes of 98.17% accuracy, 91.34% specificity, and 98.79% sensitivity [17]. The suggested SegNet-UNet model yielded improved Dice Similarity Coefficient (DSC) scores of 0.82, 0.73, and 0.68 for Whole Tumor (WT), Core Tumor (CT), and Enhanced Tumor (ET), respectively. These outcomes underscore the efficacy of combining level sets with iterative Fully Convolutional Network (FCN) architectures, as the proposed Deep Resilience Layers (DRLs) exhibited superior performance in terms of anomaly resilience, speed, and consistency in segmenting core tumors. Moreover, the integration of DRLs significantly elevated the pace of brain tumor segmentation, rendering it an efficient and effective method [18]. The researcher established a semi-supervised multi-labelling framework, denominated the Weighted Label Fusion Learning Framework (WLFS), designed for the automated segmentation of gliomas. The efficacy of the proposed technique was evaluated using datasets, including BRATS 2015, BRATS 2017, and BRATS 2019. The system architecture was compartmentalized into three segments: image preparation, graph creation, and segmentation. Labels were extended between the atlas and target images through the estimation of transmitted data. The WLFS methodology demonstrated superior segmentation accuracy, achieving 90.1%, 88.7%, and 89% for Whole Tumor (WT), Tumor Core (TC), and Enhanced Tumor (ET), respectively [19]. The researcher introduced an automated system for brain tumor segmentation based on ResNet architecture. To assess the effectiveness of the proposed methodology, simulation experiments were conducted utilizing the BRATS 2015 dataset. This shortcut connection in the ResNet model enabled the model to adopt an identity function, ensuring superior performance of the upper layers compared to the lower layers. Consequently, the suggested ResNet system demonstrated heightened segmentation accuracy, achieving 84%, 90%, and 86% for Tumor Core (TC), Enhanced Tumor (ET), and Whole Tumor (WT), respectively. Additionally, for effective feature extraction of Low-Grade Glioma (LGG) brain tumors, minor modifications to the model architecture were implemented, resulting in enhanced segmentation outcomes [20].

The application of K-means clustering and deep learning, coupled with synthetic data augmentation for classification, has been demonstrated as an effective approach for brain tumor segmentation. In the conducted experimentation, the proposed methodology underwent evaluation utilizing the BraTS 2015 dataset. Specifically, the segmentation phase involved K-means clustering, while the subsequent classification utilized an enhanced version of the VGG19 architecture. Introducing the concept of synthetic data augmentation proved pivotal, expanding the dataset available for classifier training and consequently enhancing accuracy. Through

these strategies, the suggested method achieved an impressive overall accuracy of 94%. Nevertheless, there persists a need for tumor identification and categorization approaches in MR images that are not only specific but also efficient and reliable [21]. A Multiscale 3D U-Net architecture employs multiple U-net blocks to capture extensive spatial information across varying resolutions, as demonstrated. Subsequently, feature maps are unsampled at different resolutions to comprehensively extract and process optimized features. To mitigate computational costs, temporal, and spatial complexities, 3D depth-wise separable convolution is employed on the BraTS 2015 testing set. The proposed multiscale 3D U-Net achieves an improved Dice Similarity Coefficient (DSC) of 0.85, 0.72, and 0.61 for Whole Tumor (WT), Tumor Core (TC), and Enhanced Tumor (ET), respectively [22]. The researcher devised an innovative brain tumor segmentation system implementing a Multi-Inception-UNET (MI-U-NET) architecture. The system comprises two pivotal components aimed at enhancing the precision of tumor segmentation. Firstly, a Convolutional Neural Network (CNN) was harnessed to classify slices as either tumorous or non-tumorous, effectively mitigating false positives within the system. Secondly, the MI-UNET architecture, a novel design, was utilized for segmenting the tumorous slices. MI-UNET extends the baseline UNET architecture by incorporating inception modules at each level, thereby amplifying the scalability and representation capability of the UNET model and resulting in more accurate tumor segmentation outcomes. The training of the MI-UNET model involves the application of a weighted dice loss function. Remarkably, the proposed MI-UNET achieved a superior accuracy of 94%, incorporating the use of data augmentation [22]. The proposed architectural framework of the system is formulated with metadata learning across multiple layers, seamlessly integrated with a Convolutional Neural Network (CNN) layer to ensure the delivery of precise information. A metadata-based vector encoding strategy is employed, utilizing sparse coding for additional dimension estimation. To uphold the supervised data in a geometric format, atoms of neighboring limitations are structured based on a meticulously designed k-neighboured network. The resultant system demonstrates a significant robustness and subjectivity in terms of classification. Employing two distinct datasets, namely BRATS and REMBRANDT, the brain MRI classification technique proposed in this system exhibits greater efficiency compared to other existing methodologies [23].

The presented system attains remarkable detection accuracy, boasting an overall accuracy rate of 99.53%, along with an 82.95% accuracy in the binary classification task. These outcomes signify the system's adeptness in precisely discerning epileptic seizures from EEG signals. Moreover, the proposed system exhibits a superior performance contrasted with other established techniques like K-nearest Neighbor (KNN), support vector machine (SVM), and decision tree (DT) in terms of accuracy. Notably, the evaluation and comparison of the proposed system were conducted using a publicly accessible database of epileptic seizures images. In summary, the hybrid CNN-BiLSTM architecture proposed demonstrates considerable potential for advancing the detection and classification of epileptic seizures from EEG signals, suggesting possible enhancements in the efficiency and accuracy of early-stage diagnosis and treatment for these disorders [24]. The identification of boundary edge pixels involves the application of Kirsch's edge detectors, followed by the implementation of the contrast adaptive histogram equalization method on the detected edge pixels. Subsequently, the Ridgelet transform is employed on the enhanced brain image to acquire Ridgelet multi-resolution coefficients. Features are then derived from these transformed coefficients, and Principal Component Analysis (PCA) is applied to optimize these features. The optimized features undergo classification into Glioma or non-Glioma brain images using the Co-Active Adaptive Neuro Fuzzy Expert System (CANFES) classifier. The proposed method, incorporating PCA and the CANFES classification approach, achieves noteworthy performance metrics, including

97.6% sensitivity (Se), 98.56% specificity (Sp), 98.73% accuracy (Acc), 98.85% precision (Pr), 98.11% false positive rate (FPR), and 98.185% false negative rate (FNR). These results surpass the performance of the Glioma brain tumor detection method using only the CANFES classification approach [25]. The study presents a filter-based hybrid feature selection technique based on chi-square, Relief-F, and mutual information to address the issue of dimensionality reduction in data mining. The technique assigns a score to each feature and sets a threshold value for a critical subset, reducing execution time and memory. The technique's efficacy was tested experimentally and intellectually [26]. This research aims to improve epileptic seizure identification using deep learning models. Nine architectures were trained using EEG records from participants. The Conv1D+LSTM architecture, Bidirectional LSTM, and Gated Recurrent Unit models scored well, with 0.993 effective test accuracy. Standard scaling outperformed MinMax scaling in BiLSTM and GRU accuracy. Understanding feature scaling, PCA, and feature selection affects deep learning architectures differently, improving patient outcomes and quality of life [27]. Combination therapy is a crucial cancer treatment technique, combining anti-cancer drugs to improve effectiveness and overcome multidrug resistance. This research uses machine learning to categorize and forecast cancer medication combinations, showing that combinations like Gemcitabine, MK-8776, and AZD1775 synergize against various malignancies, aiding in the development of better multi-drug regimens [28]. The study tested two Medical Concept Normalization models, MCN-BERT and BiLSTM, for automated illness prediction from symptoms using language models and deep learning. Dataset-1 and Dataset-2 were used, with the AdamP-optimized MCN-BERT model achieving 99.58% and 96.15% accuracy, respectively, while the BiLSTM model achieved 97.08% and 94.15% accuracy [29]. Incontinence, a condition characterized by uncontrolled urine leaking, can indicate pelvic floor muscle dysfunction (PFM) and pelvic tilt and lumbar angle. Traditional methods often neglect other core muscles, leading to pelvic floor dysfunction. Proposed study used decision tree, SVM, random forest, and AdaBoost models to predict core muscle activity in multiparous women with FSD, with AdaBoost achieving the highest performance [30]. Monkeypox, a rare viral disease, can cause severe skin lesions and rashes. Identifying the disease can be challenging, especially in resource-limited situations. A study suggests using CNNs to classify lesions, and the Grey Wolf Optimizer (GWO) method improved accuracy, precision, recall, F1-score, and AUC, resulting in 95.3% accuracy [31]. Author resented ML-based prediction frameworks for Hepatitis C Virus in Egyptian healthcare workers using real-world data from the National Liver Institute at Menoufiya University. The framework's robustness and dependability were tested in two scenarios: without feature selection and with sequential forward selection (SFS). The model's accuracy was improved after SFS selection, with the RF classifier learning in 0.58 seconds and 94.06% accuracy. Tweaking the RF classifier hyperparameter values improved classification accuracy to 94.88% with four features [32]. Author introduced a median filter preprocessing method to improve content-based picture retrieval accuracy and reliability. It extracts Fourier and circularity descriptors, invariant color properties, and uses multiple ant colony optimization for greedy deep Boltzmann machine classifier learning. The method outperforms a priori classification algorithms by 25% [33]. Author presented a fully automated approach using deep learning algorithms to pre-process, segment, and classify breast cancer spread intensity from patient photos. The model eliminates pectoral muscles, reduces input photos, and optimizes at each level for high accuracy. The fully automated model is compared against state-of-the-art models for comparison [34]. The study explores Content-Based Image Retrieval (CBIR) using Xception, MobileNet, and Inception deep learning architectures. The models achieved high accuracy and precision, while MobileNet performed well with 87.125% accuracy and 88% precision. The study emphasizes picture database security, using dual-layer encryption for data protection, demonstrating the model's ability to balance high retrieval

**Table 1. Comparison of state-of-the-art method.**

| Author | Year | Dataset | Method | Limitation |
|---|---|---|---|---|
| [15] | 2022 | BRATS 2015 | CNN-SVM | The scalability and generalizability of the methodology could be limited |
| [16] | 2022 | BraTS 2017 | Hybrid-DANet | The model's generalizability may be restricted if trained on a specific dataset, like the BraTS 2019 challenge Validation Dataset, as it may not perform well across different sources or protocols. |
| [17] | 2023 | REMBRANDT | CRNN | The model's performance on the trained data may be excellent, but its accuracy may decrease when encountering new, unseen data, affecting its generalizability and real-world applicability. |
| [18] | 2021 | REMBRANDT | UNET and LSTM | The model's real-time segmentation capabilities may be limited due to its impracticality during medical procedures or on devices with limited processing power. |
| [19] | 2021 | BRATS 2015 | DNN | The model's segmentation decisions may lack interpretability, a concern in critical medical applications like brain tumor segmentation, where understanding the rationale is crucial for informed medical decisions. |
| [20] | 2022 | BRATS 2015 | U-Net architecture | The model's performance may be impacted by the inherent biases in the training data when applied to diverse demographic groups. |
| [21] | 2021 | BRATS 2015 | VGG19 | The tumor localization stage's accuracy directly impacts the segmentation stage, as inaccuracy can lead to the inclusion of healthy tissue or miss parts of the tumor. |
| [22] | 2022 | BRATS, REMBRANDT | CDBLNL | K-means, a basic clustering algorithm, may not accurately distinguish tumor regions from healthy brain tissues due to its inability to learn complex image features. |
| [23] | 2023 | REMBRANDT | Enhanced fuzzy c-means clustering | CNN-BiLSTM frameworks require large, diverse datasets for optimal performance, while limited or lack of diversity in seizure types, patient demographics, or EEG recording conditions may affect detection accuracy. |
| [24] | 2023 | UCI epileptic seizure | CNN-BiLSTM | CNN biases can lead to inaccurate results for certain patient demographics or tumor types, making it challenging to identify and address potential biases without understanding its decision-making process. |
| [25] | 2020 | Kaggle | CANFES | The deep learning model's high accuracy in tumor grade classification on a specific MRI dataset may decrease when encountering new unseen data, limiting its practical application. |

accuracy with strict security [35]. Edge detection is crucial for image segmentation, industrial product defect detection, medical image processing, and object identification. This paper presents a unique image processing technology that automates industrial item dimensioning using fuzzy entropy, quick fuzzy edge detection, and Freeman Chain Codes. This technique efficiently identifies edges, corners, and flaws, with lower computational costs [36]. Author presented a image retrieval procedure using a Nonlinear Support Vector Machine Convolutional Neural Network (NSVM-CNN), which involves data preparation, feature extraction, and optimization. Gaussian filtering, GLCM, and k-means clustering are used to eliminate unreliable data. The NSVM-CNN model is trained to identify and retrieve relevant images, outperforming existing techniques with higher accuracy rates [37]. Author utilized an image processing technology to automate the quantification of industrial object dimensions. It utilizes fuzzy entropy for image enhancement, a rapid edge detection method, and Freeman Chain Codes (FCC) for key corner locations. This straightforward approach effectively identifies the size, shape, and defects of objects, facilitating the evaluation of products for acceptance or rejection. The method has significantly lower computational cost compared to related works [38]. Table 1 illustrates the state-of-the-art method comparison.

## 3. Material and methods

### 3.1 Material

The BRATS 2020 dataset represents a significant leap in enhancing brain tumor analysis. With an expanded collection featuring 76 cases of low-grade gliomas (LGG) and 259 cases of high-grade gliomas (HGG), it provides a more comprehensive foundation for evaluating the efficacy of brain tumor segmentation methods. Beyond these advancements, BRATS 2020 introduces a

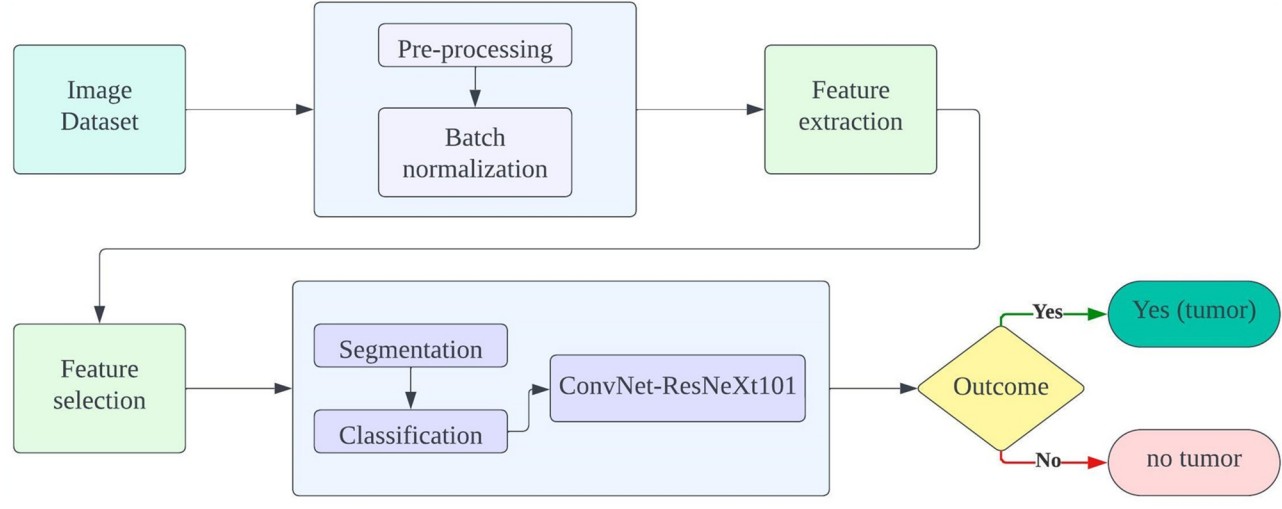

**Fig 1. Proposed ResNeXt101 operational flow.**

groundbreaking challenge: predicting the overall survival time of patients by examining their MRI scans [39]. This innovative addition pushes participants to develop models that go beyond tumor segmentation, delving into the realm of prognostic estimations and treatment planning to potentially improve patient outcomes. (https://www.kaggle.com/datasets/awsaf49/brats2020-training-data).

## 3.2 Methods

This study introduces an innovative approach to enhance the classification of brain tumors by combining batch normalization with transfer learning-based classification. The effectiveness of the proposed system is assessed using the BRATS 2020 dataset. Initially, batch normalization is applied to the images, and the AlexNet model is employed to extract features. Subsequently, an AWO algorithm is utilized for the purposeful selection of features. To achieve segmentation and improve classification, a novel method named CNN with ResNeXt101 is introduced. The visual representation of the proposed model is depicted in Fig 1.

**3.2.1 Data pre-processing.** Batch normalization, a technique that standardizes input data and accelerates neural network training, serves as the chosen preprocessing step in this experiment. Batch refers to the collection of input data, and the normalization process occurs in batches. Normalization is a tool that transforms numerical data to a consistent scale while maintaining its structure. In essence, it is the process of transforming the data to have a mean value of 0 and a standard deviation of 1. Batch normalization enhances the speed and stability of neural networks by introducing additional layers to the DNN architecture. The newly introduced layer performs standardization and normalization operations by considering the values of the preceding layer. The mean and standard deviation of the hidden activation are determined by employing batch input from layer $m$, following Eqs (1) and (2),

$$\mu = \frac{1}{n}\sum g_j \tag{1}$$

$$\sigma = \left[\frac{1}{n}\sum (g_j - \mu)^2\right]^{1/2} \tag{2}$$

where, $\mu$ refers mean, $\sigma$ refers standard deviation, $n$ refers number of neurons at layer $g$.

By applying the values obtained from Eqs (1) and (2), the hidden activations are normalized. This normalization involves subtracting the mean value from each input and then dividing by the sum of the standard deviation and a small smoothing term ($\tau$). The smoothing term, calculated using Eq (3), safeguards numerical stability by preventing division by zero.

$$g_{j(norm)} = \frac{(g_j - \mu)}{\sigma + \tau} \tag{3}$$

Subsequently, the normalized images are directed to the subsequent feature extraction process to extract pertinent features from these images.

**3.2.2 Feature extraction.** AlexNet, a groundbreaking deep CNN architecture, holds a pivotal position in deep learning (DL) for computer vision applications. In this study, we leverage AlexNet to extract meaningful features from pre-processed data. This architecture comprises five convolutional layers, three max-pooling layers, two normalization layers, two fully connected layers, and a single softmax layer. The activation function was initially employed as a model enhancement and subsequently integrated into the neural network for efficient evaluation. The AlexNet model, equipped with 96 neurons, underwent training for 100 epochs while incorporating a dropout rate of 0.5. Conventional activation functions include the arctan function, hyperbolic tangent (tanh) function, and the logistic function. However, deep learning models often face the vanishing gradient problem, arising from the presence of extremely large gradient values when input data approaches zero. To address this issue, we employ the rectified linear unit (ReLU) activation function, as defined in Eq (4).

$$ReLU(y) = maximum(y, 0) \tag{4}$$

For input data exceeding zero, the ReLU gradient is updated to 1. In deep networks, ReLU outperforms the Tanh unit in terms of convergence rate, accelerating the training process. Pooling layers reduce the feature map of a neighboring pixel group, employing various techniques to generate a value. Max pooling, for instance, produces the maximum value within each 2×2 block, while a 4×4 feature map is used to minimize feature dimensions. However, AlexNet's feature extraction process is fixed and does not adapt to the specific dataset. Therefore, an Adaptive Whale Optimization (AWO) algorithm is employed for feature selection. AWO utilizes adaptive search mechanisms that enable it to tailor the feature selection process based on the dataset's characteristics.

**3.2.3 Feature selection.** The application of the Adaptive Whale Optimization (AWO) algorithm in the realm of feature selection from Magnetic Resonance Imaging (MRI) data represents a strategic leap in optimizing the efficiency of this crucial process. Drawing inspiration from the intricate foraging behavior exhibited by whales in their natural habitats, AWO endeavours to mimic and leverage these adaptive strategies to refine the feature selection mechanism. Through the judicious use of AWO on MRI scans, the model undergoes a transformative improvement in performance. This enhancement is realized by the careful curation and selection of the most relevant features, a process that inherently contributes to the reduction of noise and the exclusion of extraneous, irrelevant information. This focused feature selection ensures that the model is trained on the most salient aspects of the MRI data, consequently refining its predictive capabilities and minimizing the risk of overfitting. The

motivation behind choosing AWO over conventional whale optimization algorithms lies in the systematic addressal of their inherent limitations. Common issues like slow convergence speed, inadequacies in global optimization, and a proclivity for falling into local optimization traps prompted the adoption of AWO. This research recognizes AWO as a more sophisticated and adaptive alternative, capable of navigating the complexities of feature selection with greater efficacy. The AWO approach unfolds in three distinct stages, each mirroring a facet of the whales' foraging behavior: Prey Search: Analogous to the whales' initial scouting for potential food sources, this stage involves an exploratory phase in feature space. Encircling Prey: Reflecting the strategic encircling observed in whales as they close in on their target, this phase narrows down the selection, homing in on the most promising features. Bubble-Net Feeding: Drawing inspiration from the collaborative bubble-net feeding behavior, this stage finalizes the feature selection, ensuring a comprehensive and refined set of features to optimize the model's predictive prowess. In essence, the integration of AWO in feature selection not only addresses the shortcomings of traditional whale optimization algorithms but also adds a layer of sophistication inspired by nature's own optimization strategies. Fig 2 depicts the work flow of proposed brain tumor classification model.

**3.2.4 Image segmentation.** Image segmentation, a cornerstone of computer vision, holds immense importance in deconstructing complex visual information into meaningful segments, enabling a more detailed and insightful analysis. This research delves into the application of advanced techniques for image segmentation, particularly ConvNet and the ResNeXt101 architecture. ConvNet, widely acclaimed for its effectiveness in visual recognition tasks, and ResNeXt101, a cutting-edge deep neural network architecture, have been meticulously selected for their exceptional capabilities in both image segmentation and classification. These models elevate the sophistication of the segmentation process, demonstrating an unparalleled ability to discern intricate patterns, shapes, and structures within input images. The utilization of ConvNet and ResNeXt101 in this research stems from their ability to transcend mere pixel-level segmentation. These models excel in comprehending contextual relationships within images, enabling them to distinguish objects, boundaries, and textures with remarkable precision. Their deep and hierarchical architectures facilitate the extraction of intricate features, contributing to the generation of finely segmented regions in the input images. The decision to employ ConvNet and ResNeXt101 is further reinforced by their adaptability to a diverse range of image segmentation tasks. Whether dealing with medical imaging, scene understanding, or object detection, these models have exhibited a versatility that aligns with the intricate requirements of various applications. In essence, by leveraging the prowess of ConvNet and ResNeXt101 for image segmentation, this research strives not only to enhance the efficiency of analysis but also to unlock new frontiers for comprehending and interpreting visual data with a degree of detail and accuracy that was once elusive.

A Convolutional Neural Network (CNN) is a class of deep neural networks specifically designed for processing and analyzing visual data, such as images and videos. CNNs have proven to be highly effective in tasks related to computer vision, including image classification, object detection, image segmentation, and more. They are inspired by the organization and functioning of the human visual system, particularly the receptive fields and hierarchical feature representation. Convolutional Layers (Conv Layers): These layers apply convolution operations to the input data, using filters or kernels. Fig 3 depicts the ConvNet architecture. Convolution involves sliding these filters over the input to detect patterns, edges, and features at different spatial hierarchies. This operation enables the network to learn hierarchical representations. Pooling Layers: Pooling layers are responsible for down-sampling the spatial dimensions of the input. Max pooling, for example, retains the maximum value from a group of neighboring pixels, reducing the spatial resolution and computational complexity while

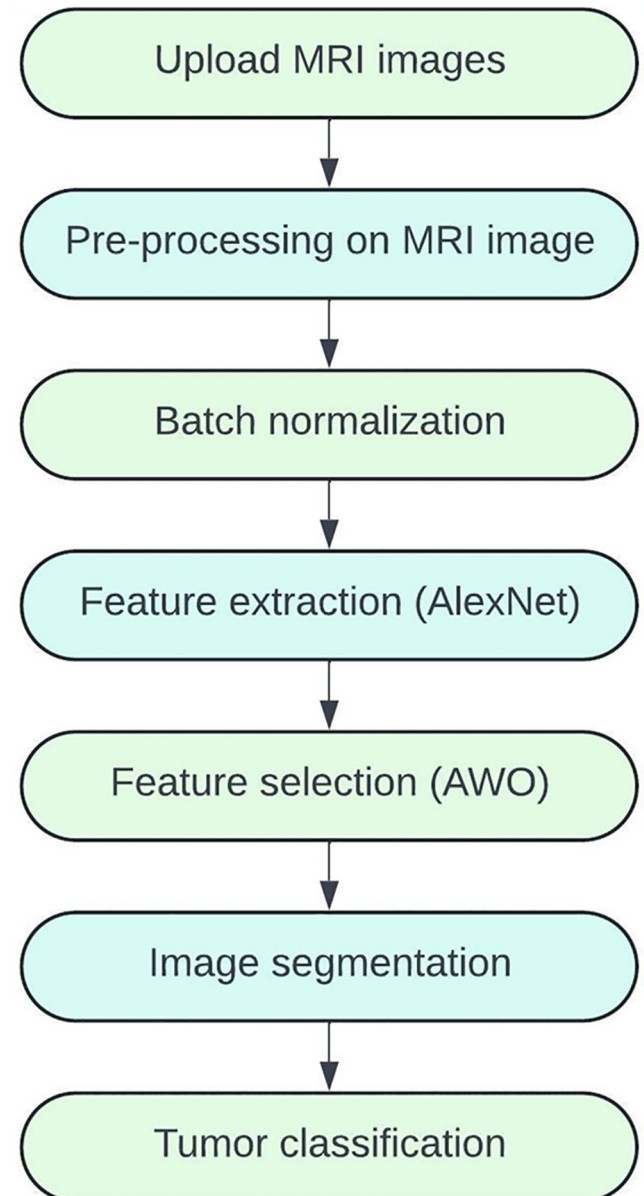

**Fig 2. The workflow of the proposed model.**

preserving important features. Activation Functions: Commonly used activation functions in CNNs include Rectified Linear Unit (ReLU), which introduces non-linearity by thresholding the input, allowing the model to learn complex patterns. Fully Connected Layers (FC Layers): These layers connect every neuron from one layer to every neuron in the next layer, leading to the final output layer. They are typically used for classification tasks. Flattening: Before the fully connected layers, the output from the convolutional and pooling layers is flattened into a vector. This vector is then used as input to the fully connected layers for decision-making. CNNs have revolutionized computer vision tasks and have achieved remarkable success in various domains, including image recognition competitions, medical image analysis, autonomous

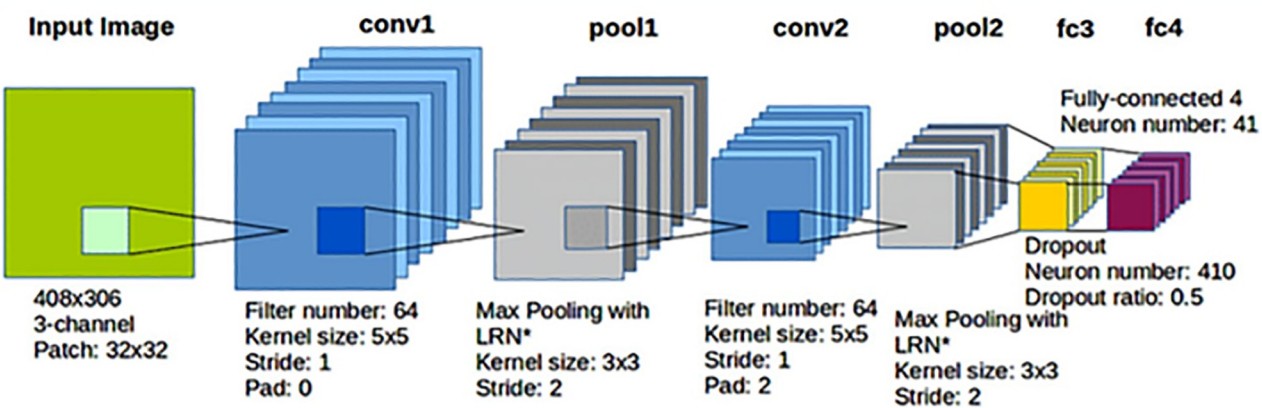

**Fig 3. Architecture of ConvNet.**

vehicles, and more. Their ability to automatically learn hierarchical representations of visual data has made them an integral part of many state-of-the-art machine learning applications.

**3.2.5 ResNeXt101.** In the dynamic landscape of deep learning, ResNeXt101 emerges as a revolutionary architectural variant within the ResNet framework. While ResNet leans on the concept of stacking recurring layers, ResNeXt101 introduces a paradigm shift by incorporating cardinal paths, thereby extending the ResNet architecture and elevating its prowess in learning intricate patterns. At the core of ResNeXt101 lies the ResNeXt101 component, a pivotal innovation that harnesses residual connections or blocks to ensure a smooth flow of information through the network. These residual connections address the notorious vanishing gradient problem, a persistent hurdle in traditional CNNs that impedes effective training and restricts the network's capacity to grasp complex patterns. By mitigating this challenge, ResNeXt101 empowers the model to navigate intricate tumor structures with heightened precision, ultimately resulting in superior segmentation accuracy. The pivotal residual block, a cornerstone of ResNeXt101, not only amplifies the network's depth but also fortifies its overall performance. Illustrated in Fig 4, the ResNeXt101 block with a cardinality of 32 exemplifies the architecture's capability to uphold complexity while achieving remarkable outcomes. This

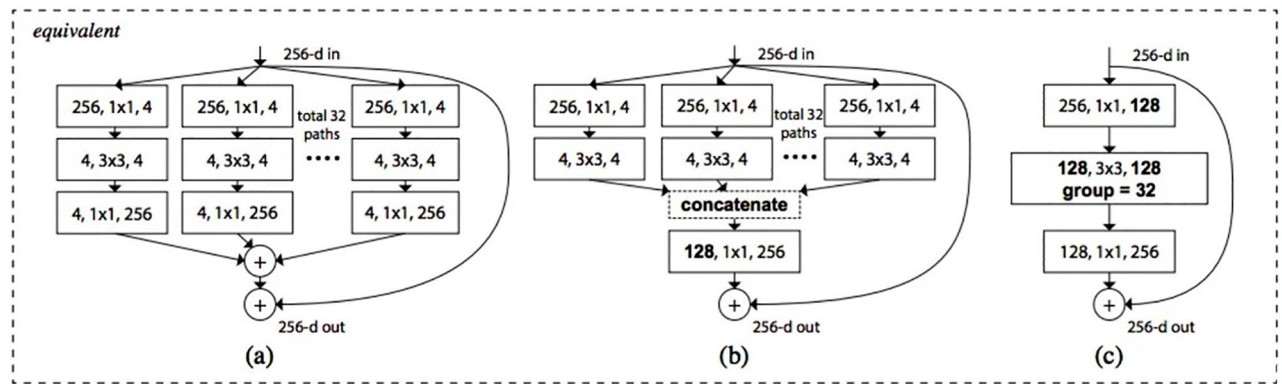

**Fig 4. ResNeXt101 (a) aggregated residual transformations; (b) implemented as early concatenation; (c) implemented as grouped convolutions.**

refined architecture played a pivotal role in ResNeXt's triumph in the ImageNet classification challenge, where its performance surpassed conventional models. Within the ResNeXt101 framework, the residual block executes the residual operation by merging the inputs with the residual block outputs. This strategic operation circumvents the main branch of the network, establishing direct connections between layers and facilitating efficient gradient propagation. The introduction of ResNeXt101 has left an indelible mark on the realm of medical image analysis, particularly in the domain of tumor segmentation. Its adeptness in handling complex tumor structures with heightened precision has not only advanced the field but has also opened up new avenues for enhancing cancer diagnosis and refining treatment planning methodologies. The success of ResNeXt101 underscores its transformative impact, positioning it as a pivotal tool in the ongoing quest for more accurate and effective medical imaging solutions.

ResNeXt101 denotes a ResNeXt101 variant with a depth of 101 layers. It is characterized by its innovative use of parallel pathways within the building blocks, known as ResNeXt101 blocks, to enhance the model's learning capabilities. Cardinality: The cardinality parameter in ResNeXt101 determines the number of parallel paths or branches within each ResNeXt101 block. In the case of ResNeXt101, the architecture leverages a high cardinality value, often denoted as 32, indicating that each block processes information through 32 parallel pathways. This allows the model to capture diverse features and patterns, contributing to its effectiveness in complex tasks. Residual Connections: Similar to ResNet, ResNeXt101 incorporates residual connections in its architecture. These connections enable the efficient flow of information through the network by providing shortcut paths for the gradient during training. This helps alleviate the vanishing gradient problem, allowing for the training of very deep neural networks. Hierarchical Feature Learning: The deep architecture of ResNeXt101 facilitates hierarchical feature learning, enabling the model to automatically learn and extract intricate features at different levels of abstraction. This hierarchical representation is crucial for tasks such as image classification, where features at various scales contribute to the understanding of complex visual patterns. Image Classification: ResNeXt101, with its depth and advanced architecture, is often employed for image classification tasks. It has demonstrated state-of-the-art performance on benchmark datasets like ImageNet, showcasing its ability to accurately classify images into multiple categories. Versatility: While ResNeXt101 is frequently used for image classification, its versatility extends to other computer vision tasks, including object detection, segmentation, and more. The architectural features that contribute to its success in image classification also make it applicable to a wide range of visual recognition tasks. Let's take a closer look at Fig 4b, which shows a key part of ResNeXt101 called the "residual block." Think of it like a special tool in the ResNeXt101 system that helps it understand pictures better. Table 2 represents the parameters used in the proposed model.

This tool is really good at finding important details in pictures, like shapes and patterns, by using different techniques. The ResNeXt101 system, kind of like a smart picture detective, has a special way of combining these techniques. Unlike some other methods that use these techniques one after the other, ResNeXt101 mixes them up to work together better. This makes it even more effective at understanding pictures. One cool thing about ResNeXt101 is that it can change its way of working based on what it's looking at. If it needs to pay more attention to one aspect or another, it can do that. It's like having a tool that can adjust itself to solve different kinds of picture puzzles. In Fig 3, you can see three versions of this special tool in ResNeXt101. Each version is like a different setting on a camera—it's adjusted for different situations. This flexibility helps ResNeXt101 adapt to lots of different types of pictures, making it really handy for different jobs. Experts have noticed that ResNeXt101 is easier to teach compared to some other systems, showing that it's versatile and works well with lots of different

**Table 2. Proposed model tuned hyperparameters.**

| Stage | Output | ResNeXt101 |
|---|---|---|
| conv_1 | 112×112 | 7×7, 64 stride 2 |
| conv_2 | 56×56 | 3x3 max_pool, stride 2 |
| | | $\begin{bmatrix} 1 \times 1, & 128 \\ 3 \times 3, & 128, C = 32 \\ 1 \times 1, & 256 \end{bmatrix} \times 3$ |
| conv_3 | 28×28 | $\begin{bmatrix} 1 \times 1, & 256 \\ 3 \times 3, & 256, C = 32 \\ 1 \times 1, & 512 \end{bmatrix} \times 4$ |
| conv_4 | 14×14 | $\begin{bmatrix} 1 \times 1, & 512 \\ 3 \times 3, & 512, C = 32 \\ 1 \times 1, & 1024 \end{bmatrix} \times 6$ |
| conv_5 | 7×7 | $\begin{bmatrix} 1 \times 1, & 1024 \\ 3 \times 3, & 1024, C = 32 \\ 1 \times 1, & 2048 \end{bmatrix} \times 3$ |
| | 1×1 | global avg_pool 1000-d fully connected, softmax |
| #parameters | | 25×106 |
| FLOPs | | 4.2×109 |

picture collections. After ResNeXt101 looks at and understands the pictures, it then sorts them into categories like a brain tumor classifier. This whole process, from understanding the pictures to categorizing them, shows how ResNeXt101 is really good at solving complicated problems in medical image analysis. Table 3 illustrates the ResNeXt101 training outcomes, and ResNeXt-101 are less error-prone when the cardinality is high.

**3.2.6 Image classification.** The classification stage evaluates the performance of the segmented feature set using a chosen classification method. Each classifier undergoes a training and testing phase, involving the division of the dataset into training and validation sets. This study employs a deep learning-based CNN and a transfer learning-based ResNeXt101 classifier for effective brain tumor classification. The segmented features are initially fed into the ConvNet-ResNeXt101 model, and the classifier subsequently passes the images through its layers. Finally, the convolutional layers classify the image's features by scanning the input image with multiple filters. Leveraging the robust neural network architecture of ResNeXt101, the process of image classification involves accurately sorting images into predefined categories or labels. ResNeXt101, renowned for its adeptness in handling intricate visual recognition tasks, emerges as an ideal choice for tackling the challenges posed by image classification. To initiate this process: Assemble and organize a labelled dataset of images for training the model, ensuring its diversity and representation of the desired recognition classes. Further, divide the dataset into

**Table 3. ResNeXt101 training and results.**

| Model | Parameter Setting | Top-1 error (%) |
|---|---|---|
| ResNeXt101 | 2×40d | 21.7 |
| | 4×24d | 21.4 |
| | 8×14d | 21.3 |
| | 32×4d | 21.2 |

training and validation sets to gauge the model's effectiveness. Establish the model architecture by initializing a ResNeXt101 model using popular deep learning frameworks such as Tensor-Flow or PyTorch. Incorporate the pre-trained weights of ResNeXt101, often fine-tuned on extensive datasets like ImageNet, capturing foundational features that can be adapted for specific classification objectives. Optionally, engage in fine-tuning or transfer learning. Fine-tuning involves adjusting the model's weights based on your dataset, enhancing its specialization. Alternatively, opt for transfer learning by retaining the pre-trained weights and introducing a new classification layer tailored to your specific task. Implement data augmentation techniques during training to artificially diversify the dataset. This augmentation enhances the model's ability to generalize to novel, unseen images. Train the ResNeXt101 model on the prepared dataset, refining its weights based on the input images and their associated labels. Regularly assess the model's performance on the validation set to prevent overfitting. Evaluate the trained model on an independent test dataset, gauging its capacity to generalize to new and unseen images. Utilize metrics such as accuracy, precision, recall, and F1 score to comprehensively measure the model's efficacy. Deploy the trained ResNeXt101 model for real-world image classification tasks. Introduce new images to the model, allowing it to predict their respective categories. Optionally, fine-tune the model based on its real-world performance, further refining its accuracy over time to align with specific application needs.

## 4. Experimental result and discussion

This section delves into the results and analysis of the implemented and simulated ConvNet-ResNeXt101 model, leveraging the power of Python software and a robust computer equipped with 16GB RAM, an INTEL i7 processor, Windows 11 operating system, an 8GB GPU, and a spacious 1TB HDD. The effectiveness of the proposed model is meticulously evaluated using the established performance metrics outlined in Eqs (5–10). In this section, an in-depth exploration is undertaken to assess the segmentation performance of the proposed ConvNet-ResNeXt101 model. This evaluation involves a comparative analysis with well-established counterparts such as CNN, VGG16, and ResNet51. The focus of the scrutiny lies in unravelling the intricacies of sensitivity and specificity across Central Tumor (CT), Entire Tumor (ET), and Optimized Tumor (OT) classes, meticulously presented in Table 4. Moving beyond conventional metrics, Table 3 sheds light on the models' performance through the lens of the dice similarity coefficient (DSC), providing a holistic perspective on their capabilities. Sensitivity, also recognized as the true positive rate (TP), serves as a metric for gauging the model's proficiency in accurately identifying genuine positives. In the context of tumor detection, it manifests how effectively the model discerns authentic tumor regions. Conversely, specificity, characterized as the true negative rate (TN), elucidates the model's precision in correctly identifying non-tumor regions. This metric accentuates the model's adeptness in avoiding the

**Table 4. Proposed model sensitivity outcome.**

| Models | Sensitivity (%) | | |
|---|---|---|---|
| | Central tumor | Optimized tumor | Entire tumor |
| CNN | 76.69 | 70.68 | 86.74 |
| VGG16 | 84.66 | 76.66 | 90.63 |
| VGG19 | 89.63 | 80.65 | 92.63 |
| ResNet51 | 91.6 | 86.63 | 92.59 |
| InceptionNet | 96.59 | 96.6 | 98.6 |
| ConvNet+ResNeXt101 | 97.63 | 97.64 | 98.61 |

misclassification of normal tissue as tumors. The DICE Similarity Coefficient (DSC) takes the assessment a step further by quantifying the spatial overlap between the predicted segmentation and the ground truth. A DSC value of 1 signifies perfect overlap, while 0 indicates no spatial agreement. Consequently, DSC offers a nuanced evaluation of the model's capacity to precisely delineate tumor boundaries. Tables 3 and 4 meticulously showcase the performance metrics of ConvNet-ResNeXt101 alongside other models, emphasizing DSC, sensitivity, and specificity, respectively. The interpretation of these metrics provides invaluable insights into the strengths and weaknesses of each model, facilitating the selection of the most appropriate model for specific applications. While quantitative metrics like DSC and sensitivity/specificity contribute significant information, it is imperative to acknowledge that they offer only a partial representation of the overall performance. The importance of visual assessment of segmentation results cannot be overstated, as it enables the identification of potential artifacts or misclassifications that may elude the scrutiny of metrics alone. A comprehensive understanding of ConvNet-ResNeXt101 and other models emerges from the synthesis of both quantitative and qualitative considerations, paving the way for informed decision-making in the realm of medical image segmentation tasks.

$$Sensitivity\ (Se) = \frac{TP}{TP + FN} \times 100 \tag{5}$$

$$Specificity\ (Sp) = \frac{TN}{TN + FP} \times 100 \tag{6}$$

$$Accuracy\ (Acc) = \frac{TP + TN}{TP + TN + FP + FN} \times 100 \tag{7}$$

$$Precision\ (\text{Pr}) = \frac{TP}{TP + FP} \times 100 \tag{8}$$

$$F\ measure = \frac{2PR}{P + R} \times 100 \tag{9}$$

$$Dice\ similarity\ coefficient\ (DSC) = \frac{2TP}{FN + FP + 2TP} \times 100 \tag{10}$$

In Table 4, the sensitivity results across different tumor types for various models provide insights into their performance. The Convolutional Neural Network (CNN) demonstrates sensitivity scores of 76.69%, 70.68%, and 86.74% for central, optimized, and entire tumors, respectively. VGG16 exhibits improved sensitivity, particularly in central and entire tumors, with scores of 84.66% and 90.63%. VGG19 further enhances sensitivity, achieving scores of 89.63%, 80.65%, and 92.63% for central, optimized, and entire tumors, respectively. ResNet51 continues the trend of improvement, showcasing sensitivity scores of 91.6%, 86.63%, and 92.59% for central, optimized, and entire tumors. InceptionNet achieves high sensitivity across all tumor types, with scores of 96.59%, 96.6%, and 98.6%. Notably, ConvNet+ResNeXt101 emerges as a top performer, demonstrating outstanding sensitivity with scores of 97.63%, 97.64%, and 98.61% for central, optimized, and entire tumors, respectively. These results highlight the nuanced performance of each model in identifying specific tumor types, offering valuable information for model selection in medical image segmentation tasks. Fig 5 depicts the bar chart representation of proposed model sensitivity result.

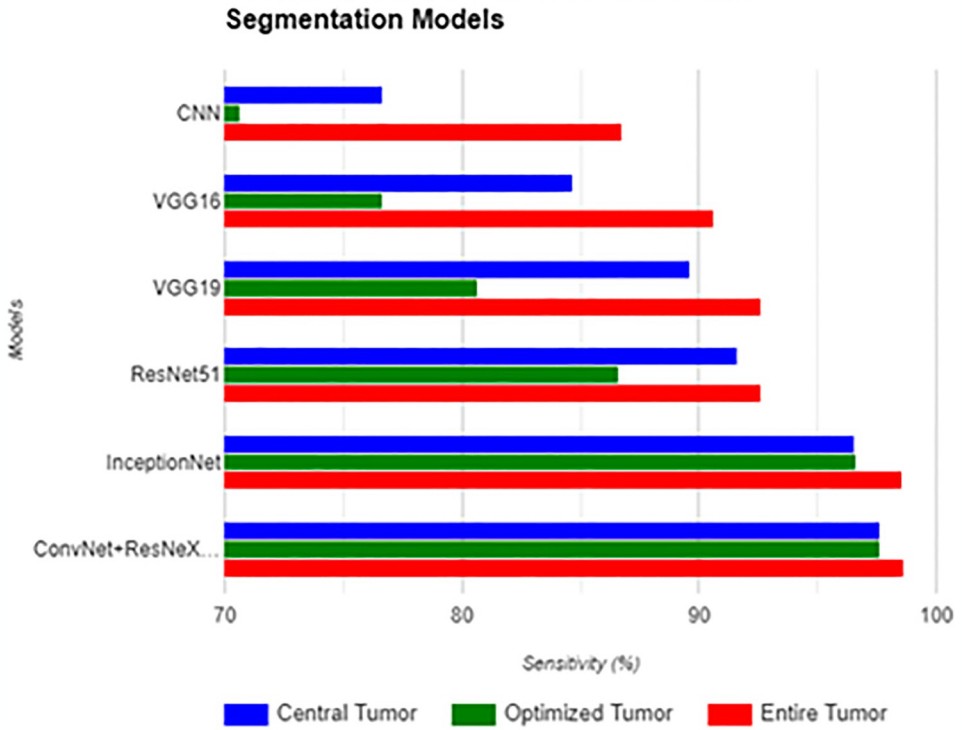

**Fig 5. Bar chart representation of proposed model sensitivity outcome.**

In Table 5, the detailed results obtained from the Sensitivity and Dice Similarity Coefficient (DSC) analyses provide a comprehensive overview of the performance of various models in tumor detection across distinct tumor types. Notably, the Convolutional Neural Network (CNN) demonstrates a strong ability to identify central tumor regions with a sensitivity of 93.69%, while VGG16 and VGG19 exhibit incremental improvements in sensitivity, showcasing enhanced performance in detecting central and optimized tumors. The ResNet51 model showcases significant sensitivity improvements across all tumor types, particularly excelling in identifying entire tumor regions with a remarkable sensitivity of 97.59%. Fig 6 depicts the proposed model sensitivity and DSC outcome.

InceptionNet consistently outperforms its predecessors, achieving high sensitivity scores across central, optimized, and entire tumor classes. Moving to DSC analysis, all models demonstrate high overall scores, indicating robust spatial agreement between predicted and actual

**Table 5. Proposed model specificity and DSC outcomes.**

| Models | Specificity (%) | | | DSC (%) | | |
|---|---|---|---|---|---|---|
| | Central tumor | Optimized tumor | Entire tumor | Central tumor | Optimized tumor | Entire tumor |
| CNN | 93.69 | 93.68 | 92.74 | 81.68 | 71.69 | 86.65 |
| VGG16 | 94.66 | 95.66 | 96.63 | 85.66 | 83.65 | 90.63 |
| VGG19 | 96.63 | 96.65 | 95.63 | 86.63 | 88.69 | 86.66 |
| ResNet51 | 97.6 | 96.63 | 97.59 | 91.64 | 92.58 | 92.69 |
| InceptionNet | 98.59 | 98.6 | 98.7 | 96.69 | 97.35 | 97.68 |
| ConvNet+ResNeXt101 | 99.89 | 99.92 | 99.78 | 99.89 | 99.92 | 99.78 |

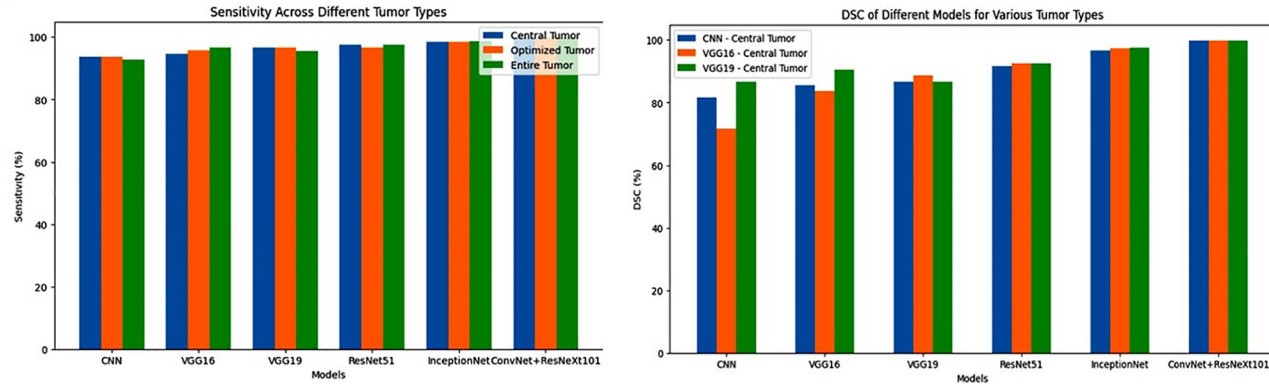

**Fig 6. Graphical illustration of sensitivity and DSC outcome of proposed model.**

tumor regions. ConvNet+ResNeXt101 stands out with the highest DSC scores, showcasing precise delineation of central tumor boundaries and exceptional spatial overlap for entire tumors. Both VGG19 and ResNet51 consistently achieve commendable DSC scores across all tumor types, reflecting their precision in capturing spatial agreement. However, a common challenge observed among all models is the relatively lower DSC scores for optimized tumor delineation, suggesting potential difficulties in accurately capturing spatial overlap in this specific tumor type. This underscores the ongoing need for optimization and refinement efforts in medical image segmentation tasks, especially concerning optimized tumor delineation. In conclusion, the results depict a progressive improvement in sensitivity and DSC scores from simpler models like CNN to more complex architectures such as ConvNet+ResNeXt101. ConvNet+ResNeXt101 emerges as the top-performing model, excelling in optimized tumor detection with the highest sensitivity and DSC scores. The challenges observed in optimized tumor delineation underscore the ongoing need for optimization and refinement efforts in the field of medical image segmentation. Fig 7 illustrates the performance metric comparison of proposed and other models. These findings contribute valuable insights to the selection and optimization of models for improved accuracy in tumor detection, aiding advancements in medical imaging applications. The comprehensive evaluation of various models for medical image segmentation reveals compelling results across key metrics. The accuracy analysis indicates a substantial improvement from simpler models, such as CNN, to more complex architectures like ConvNet+ResNeXt101. ConvNet+ResNeXt101 emerges as the top-performing model, achieving remarkable accuracy percentages of 99.27%, 98.77%, and 98.97% for Central Tumor, Optimized Tumor, and Entire Tumor, respectively. Precision metrics highlight the models' ability to make accurate positive predictions, with ConvNet+ResNeXt101 demonstrating exceptional precision, particularly evident in the optimized tumor category. F score analysis, combining precision and recall, underscores ConvNet+ResNeXt101's superior performance, with F scores reaching 98.67%, 99.07%, and 98.27% across the three tumor types. These results collectively emphasize the effectiveness of ConvNet+ResNeXt101 in medical image segmentation tasks, showcasing its potential for precise and reliable tumor region delineation. Fig 8 depicts the accuracy comparison of proposed and other conventional models.

Table 6, illustrates the classification accuracy comparison of proposed and other state-of-the-art methods. In the context of medical image segmentation, various deep learning architectures have been applied to different Brain Tumor Segmentation (BRATS) challenges. For

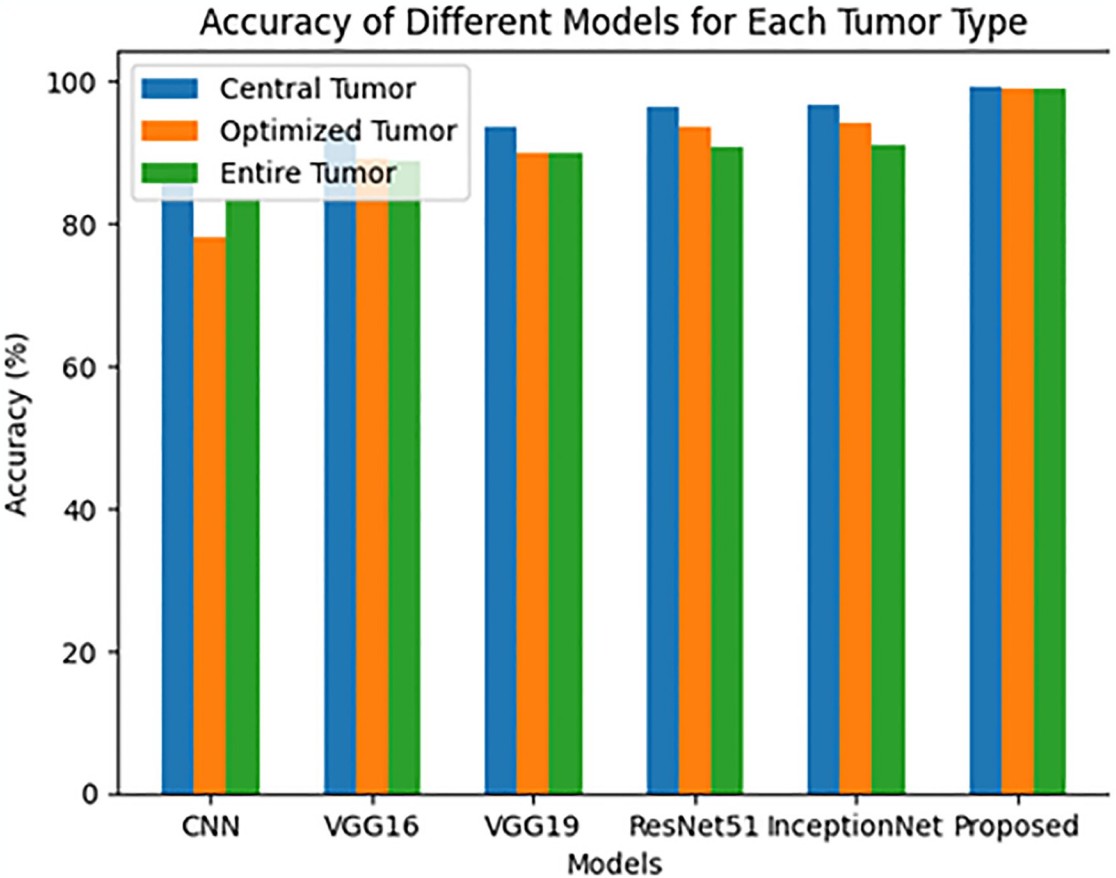

**Fig 7. Accuracy comparison of proposed and other models.**

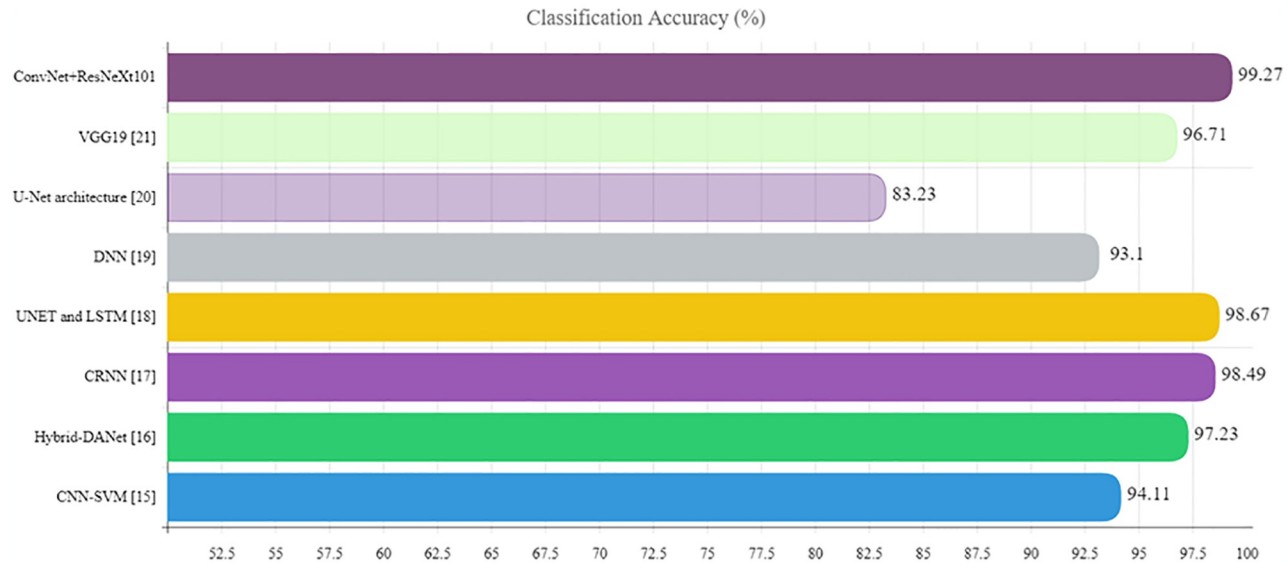

**Fig 8. Graphical illustration of accuracy comparison of proposed and other existing models.**

**Table 6. Performance metric comparison of proposed and other existing models.**

| Models | Accuracy (%) | | | Precision (%) | | | F score (%) | | | Specificity (%) | | | DSC (%) | | | Sensitivity (%) | | |
|---|---|---|---|---|---|---|---|---|---|---|---|---|---|---|---|---|---|---|
| | CT | OT | ET | CT | OT | ET | CT | OT | ET | CT | OT | ET | CT | OT | ET | CT | OT | ET |
| CNN | 86.14 | 77.95 | 83.62 | 81.6 | 70.49 | 85.34 | 91.58 | 92.25 | 95.47 | 93.69 | 93.68 | 92.74 | 81.68 | 71.69 | 86.65 | 76.69 | 70.68 | 86.74 |
| VGG16 | 92.92 | 89.02 | 88.88 | 87.89 | 79.57 | 90.93 | 94.96 | 94.99 | 97.03 | 94.66 | 95.66 | 96.63 | 85.66 | 83.65 | 90.63 | 84.66 | 76.66 | 90.63 |
| VGG19 | 93.58 | 89.91 | 89.89 | 88.91 | 79.91 | 93.17 | 95.96 | 96.47 | 97.78 | 96.63 | 96.65 | 95.63 | 86.63 | 88.69 | 86.66 | 89.63 | 80.65 | 92.63 |
| ResNet51 | 96.46 | 93.58 | 90.79 | 93.68 | 88.67 | 95.26 | 97.47 | 96.35 | 97.57 | 97.6 | 96.63 | 97.59 | 91.64 | 92.58 | 92.69 | 91.6 | 86.63 | 92.59 |
| InceptionNet | 96.78 | 94.25 | 90.91 | 93.91 | 88.92 | 95.68 | 97.79 | 96.87 | 97.69 | 98.59 | 98.6 | 98.7 | 96.69 | 97.35 | 97.68 | 96.59 | 96.6 | 98.6 |
| ConvNet+ResNeXt101 | 99.27 | 98.77 | 98.97 | 98.07 | 98.37 | 97.97 | 98.67 | 99.07 | 98.27 | 99.89 | 99.92 | 99.78 | 99.89 | 99.92 | 99.78 | 97.63 | 97.64 | 98.61 |

BRATS 2015, the CNN-SVM [15] achieved an accuracy of 94.11%, while in BRATS 2017, the Hybrid-DANet [16] demonstrated an improved performance with an accuracy of 97.23%. The REMBRANDT dataset witnessed the application of CRNN [17], yielding an accuracy of 98.49%, and UNET and LSTM [18], achieving an accuracy of 98.67%. For BRATS 2015, the DNN [19] exhibited an accuracy of 93.10%, U-Net architecture [20] achieved 83.23%, and VGG19 [21] reached an accuracy of 96.71%. In the latest BRATS 2020, ConvNet+ResNeXt101 outperformed others with an impressive accuracy of 99.27%. Table 7 represents the classification accuracy comparison of proposed and other existing models. These results underscore the effectiveness of diverse deep learning models in addressing the complexities of brain tumor segmentation across different datasets and time periods. Fig 7 depicts the graphical illustration of accuracy comparison of proposed and other existing models.

## 5. Conclusion

The accurate detection and subsequent management of brain tumors hold paramount importance in providing effective care for individuals grappling with this medical condition. This comprehensive study relies on the utilization of Magnetic Resonance Imaging (MRI) images procured from the publicly accessible BRATS 2020 dataset, a valuable resource for advancing research in brain tumor segmentation. To optimize the quality of these images, a sophisticated batch normalization technique is strategically implemented during the preprocessing phase, enhancing the overall robustness and reliability of the subsequent analysis. The feature extraction process is orchestrated through the application of the state-of-the-art AlexNet model, renowned for its proficiency in discerning intricate patterns within medical imagery. Following this, the study employs the AWO feature selection approach to meticulously identify and prioritize the most pertinent features extracted from the enhanced images. These meticulously selected features are then subjected to segmentation and classification using the proposed

**Table 7. Classification accuracy comparison of proposed and state-of-the-art methods.**

| Author | Dataset | Method | Classification Accuracy |
|---|---|---|---|
| [15] | BRATS 2015 | CNN-SVM | 94.11% |
| [16] | BraTS 2017 | Hybrid-DANet | 97.23% |
| [17] | REMBRANDT | CRNN | 98.49% |
| [18] | REMBRANDT | UNET and LSTM | 98.67% |
| [19] | BRATS 2015 | DNN | 93.10% |
| [20] | BRATS 2015 | U-Net architecture | 83.23% |
| [21] | BRATS 2015 | VGG19 | 96.71% |
| [22] | BRATS 2020 | ConvNet+ResNeXt101 | 99.27% |

ConvNet-ResNeXt101 model, an innovative architecture designed to outperform conventional counterparts. In a comparative evaluation, the ConvNet-ResNeXt101 model stands out by significantly streamlining hyperparameter requirements when juxtaposed with traditional network models. This is achieved by introducing "cardinality," an additional dimension that complements the width and depth parameters in the ResNet architecture. Experimental validation underscores the efficacy of this augmentation, showcasing a substantial enhancement in classification accuracy. The ConvNet-ResNeXt101 model achieves an impressive accuracy rate of 99.27%, eclipsing the performance of existing models, and underscores its potential as a cutting-edge solution in the realm of brain tumor segmentation. Comprehensive analyses further affirm the ConvNet-ResNeXt101 model's superiority, not only in terms of accuracy but also across critical metrics such as Dice Similarity Coefficient (DSC), Sensitivity, Specificity, and overall Accuracy. As the study concludes, future efforts will be directed toward fine-tuning hyperparameters within the classifier, aiming to achieve even more precise and reliable results. Moreover, the elapsed learning time is very law which means the speed of the learning process of the proposed framework. In the future, ConvNet+ResNeXt101 image classification will focus on refining and improving its capabilities through a variety of approaches. This includes fine-tuning hyperparameters for optimal performance on different datasets and exploring advanced techniques like data augmentation and pretraining to enhance adaptability. The model will also investigate ensemble learning methods and architectural modifications, such as adjusting cardinality and incorporating attention mechanisms, to achieve further enhancements. Future efforts will emphasize improving interpretability, addressing imbalanced data issues, and exploring domain adaptation techniques for broader applicability. Additionally, research will be directed towards model quantization and compression techniques to enable deployment on resource-constrained platforms and real-time applications. These strategies collectively outline a comprehensive plan for the ongoing development of ConvNet+-ResNeXt101, ensuring its effectiveness across diverse and dynamic contexts.

## Author Contributions

**Conceptualization:** Subathra Gunasekaran, Prabin Selvestar Mercy Bai.

**Data curation:** Hariharan Rajadurai, Basu Dev Shivahare.

**Formal analysis:** Mohd Asif Shah.

**Funding acquisition:** Mohd Asif Shah.

**Methodology:** Sandeep Kumar Mathivanan.

**Supervision:** Hariharan Rajadurai.

**Validation:** Hariharan Rajadurai, Basu Dev Shivahare.

**Visualization:** Hariharan Rajadurai, Basu Dev Shivahare.

**Writing – original draft:** Subathra Gunasekaran, Prabin Selvestar Mercy Bai.

**Writing – review & editing:** Sandeep Kumar Mathivanan.

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
