## [Decision Letter · Decision Letter 0]

8 Apr 2024

PONE-D-24-03396Automated Brain Tumor Diagnostics: Empowering Neuro-Oncology with Deep Learning-Based MRI Image AnalysisPLOS ONE

Dear Dr. Shah,

Thank you for submitting your manuscript to PLOS ONE. After careful consideration, we feel that it has merit but does not fully meet PLOS ONE’s publication criteria as it currently stands. Therefore, we invite you to submit a revised version of the manuscript that addresses the points raised during the review process.

We look forward to receiving your revised manuscript.

Kind regards,

Toqeer Mahmood, Ph.D.

Academic Editor

PLOS ONE

Journal Requirements:

Reviewers' comments:

Reviewer's Responses to Questions

**Comments to the Author**

1. Is the manuscript technically sound, and do the data support the conclusions?

Reviewer #1: Partly

Reviewer #2: Yes

Reviewer #3: Partly

2. Has the statistical analysis been performed appropriately and rigorously? 

Reviewer #1: Yes

Reviewer #2: Yes

Reviewer #3: Yes

3. Have the authors made all data underlying the findings in their manuscript fully available?

Reviewer #1: Yes

Reviewer #2: Yes

Reviewer #3: Yes

4. Is the manuscript presented in an intelligible fashion and written in standard English?

Reviewer #1: Yes

Reviewer #2: Yes

Reviewer #3: Yes

5. Review Comments to the Author

**Reviewer #1: **1. Pseudocode / Flowchart and algorithm steps need to be inserted.

2. Time spent need to be measured in the experimental results.

3. Limitation and Discussion Sections need to be inserted.

4. All metrics need to be calculated such as Recall, LossROC curves and AUC score as a table in the experimental results.

5. The parameters used for the analysis must be provided in table

6. The architecture of the proposed model must be provided

7. Address the accuracy/improvement percentages in the abstract and in the conclusion sections, as well as the significance of these results.

8. The authors need to make a clear proofread to avoid grammatical mistakes and typo errors.

9. Enhance the clarity of the Figures by improving their resolution.

10. Add future work in last section (conclusion) (if any)

11. The authors need to add recent articles in related work and update them.

12. To improve the Related Work and Introduction sections authors are recommended to review this highly related research work paper:

a) Developing an Efficient Method for Automatic Threshold Detection Based on Hybrid Feature Selection Approach

b) Optimizing epileptic seizure recognition performance with feature scaling and dropout layers

c) Harnessing machine learning to find synergistic combinations for FDA-approved cancer drugs

d) Optimizing classification of diseases through language model analysis of symptoms

e) Predicting female pelvic tilt and lumbar angle using machine learning in case of urinary incontinence and sexual dysfunction

f) Utilizing convolutional neural networks to classify monkeypox skin lesions

g) Hepatitis C Virus prediction based on machine learning framework: a real-world case study in Egypt

**Reviewer #2: **1- Explaining the researcher’s contributions in clear and sequential points

2- The references listed in Related Works are not arranged according to year of publication (from oldest to newest). In addition to clarifying the strengths and weaknesses of each research paper

3- Explain the proposed work algorithm in points before beginning to explain each point in detail, as was done in the research paper

4-Conclusions need to be supported by numerical values from the results

5-Standardize the format of references

**Reviewer #3: **Some comments for the authors to improve the quality of the article

1- Explain the abbreviation in the first use.

2- What does ResNeXt101 stand for? Usually, the ResNeXt10 algorithm is used for 1D data and 2D. How is the visualization of the image data after applying the ResNeXt10 algorithm.

3- The “Introduction” and “Related Work” sections lack of enough references. I strongly recommend that the author improve this section by adding references that support all the claims and motivation of the problem. The author may precisely and comprehensively point out the current issues and existing solutions. I suggest adding more related reference such as:

a- https://ieeexplore.ieee.org/abstract/document/10463420

b- https://ieeexplore.ieee.org/abstract/document/8876847

c- https://ijeecs.iaescore.com/index.php/IJEECS/article/view/27938

d- https://jou.jobrs.edu.iq/index.php/home/article/view/63

e- https://www.informatica.si/index.php/informatica/article/view/4840

f- https://link.springer.com/article/10.1186/s13673-019-0191-8

g- https://ieeexplore.ieee.org/abstract/document/10086271

4- Why are the results in Table 3 different from those in Table 4?

5- Does the ResNeXt10 algorithm perform before or after splitting the data? In the research methodology, the process of data splitting was not explained

6- Authors should add the computational complexities and cost for all reviewed works.

6- How authors tuned the hyperparameters?

7- Limitations and the future scope should be added with more clarity.

6. PLOS authors have the option to publish the peer review history of their article (what does this mean?). If published, this will include your full peer review and any attached files.

Reviewer #1: **Yes: **Tarek Abd El-Hafeez

Reviewer #2: No

Reviewer #3: No

---

## [Author Response · Author response to Decision Letter 0]

24 Apr 2024

Reviewer #1: 

1. Pseudocode / Flowchart and algorithm steps need to be inserted.

Response: Thank you sincerely for dedicating your valuable time. We have included the proposed model algorithm to enhance readers' comprehension.

2. Time spent need to be measured in the experimental results.

Response: Based on your valuable time and, we have incorporated the time consumption data pertaining to the outcomes obtained from the proposed model.

3. Limitation and Discussion Sections need to be inserted.

Response: Based on your valuable time and suggestion, the limitation of the state-of-the-art methods have been included in related work section.

4. All metrics need to be calculated such as Recall, LossROC curves and AUC score as a table in the experimental results.

Response: Thank you for your valuable time and suggestions. Currently, we are utilizing six metric parameters to evaluate our proposed model. We highly value your suggestions and assure you that in our future work, we will incorporate additional recommended parameters to further enhance our outcomes.

5. The parameters used for the analysis must be provided in table

Response: Based on your valuable time and suggestion all the parameter outcomes are included in the Table 5.

6. The architecture of the proposed model must be provided

Response: Thank you for your valuable comment. The proposed model architecture is shown in Figure 1. 

7. Address the accuracy/improvement percentages in the abstract and in the conclusion sections, as well as the significance of these results.

Response: Based on your valuable suggestion, we have included the accuracy outcome in abstract and conclusion section as well.

8. The authors need to make a clear proofread to avoid grammatical mistakes and typo errors.

Response: Based on your valuable time and suggestion, we carefully reviewed our article, fixing all the grammar and spelling mistakes. 

9. Enhance the clarity of the Figures by improving their resolution.

Response: Based on your valuable time and comment, we carefully checked all the figures in our article and the appropriate action has been taken.

10. Add future work in last section (conclusion) (if any)

Response: Thank you for your valuable time and comment. The future work of the proposed model has been included in the conclusion section

11. The authors need to add recent articles in related work and update them.

Response: Based on your valuable time and comment, we included the recently published articles in related work section and properly cited in reference section as well.

12. To improve the Related Work and Introduction sections authors are recommended to review this highly related research work paper:

a) Developing an Efficient Method for Automatic Threshold Detection Based on Hybrid Feature Selection Approach

b) Optimizing epileptic seizure recognition performance with feature scaling and dropout layers

c) Harnessing machine learning to find synergistic combinations for FDA-approved cancer drugs

d) Optimizing classification of diseases through language model analysis of symptoms

e) Predicting female pelvic tilt and lumbar angle using machine learning in case of urinary incontinence and sexual dysfunction

f) Utilizing convolutional neural networks to classify monkeypox skin lesions

g) Hepatitis C Virus prediction based on machine learning framework: a real-world case study in Egypt

Response: Based on your valuable time and suggestion, the above forementioned articles are included in related work section and properly cited in reference section as well.

Reviewer #2: 

1- Explaining the researcher’s contributions in clear and sequential points.

Response: Based on your valuable time and suggestion, we have included the research contribution using bullet points for better understanding of the readers.

2- The references listed in Related Works are not arranged according to year of publication (from oldest to newest). In addition to clarifying the strengths and weaknesses of each research paper.

Response: Thank you for your valuable time and comment. We have arranged the reference list based on your suggestion and included the limitation of the existing models in table 1.

3- Explain the proposed work algorithm in points before beginning to explain each point in detail, as was done in the research paper.

Response: Thank you for dedicating your time to provide feedback. We have incorporated the proposed model workflow, represented as Figure 2, to enhance the readers' comprehension of the process.

4-Conclusions need to be supported by numerical values from the results

Response: Thanks to your valuable input, we've incorporated the specific numerical value of the result, ensuring clarity and accuracy in our presentation.

5-Standardize the format of references

Response: Thanks to your valuable feedback, we have revised the reference format to align with the structure specified by the journal guidelines.

Reviewer #3: 

Some comments for the authors to improve the quality of the article

1- Explain the abbreviation in the first use.

Response: Based on your valuable time and comment, we have carefully checked the abbreviation that can be abbreviated only once.

2- What does ResNeXt101 stand for? Usually, the ResNeXt10 algorithm is used for 1D data and 2D. How is the visualization of the image data after applying the ResNeXt10 algorithm.

Response: Thank you for your valuable time and comment. ResNeXt101 stands for "Residual Next with 101 layers". It's a convolutional neural network (CNN) architecture that incorporates a large number of layers (101 in this case) along with residual connections to effectively learn features from complex image data. While ResNeXt10 might be more commonly associated with 1D or 2D data, ResNeXt101 is specifically designed and widely used for processing high-dimensional image data, such as the MRI images used in brain tumor classification tasks.

After applying the ResNeXt101 algorithm to image data, the visualization often reveals enhanced feature representation and finer details, thanks to the network's ability to capture hierarchical patterns and relationships within the images. This results in more accurate and reliable segmentation and classification outcomes, providing researchers and practitioners with valuable insights into the underlying structures and characteristics of the image data.

3- The “Introduction” and “Related Work” sections lack of enough references. I strongly recommend that the author improve this section by adding references that support all the claims and motivation of the problem. The author may precisely and comprehensively point out the current issues and existing solutions. I suggest adding more related reference such as:

a- https://ieeexplore.ieee.org/abstract/document/10463420

b- https://ieeexplore.ieee.org/abstract/document/8876847

c- https://ijeecs.iaescore.com/index.php/IJEECS/article/view/27938

d- https://jou.jobrs.edu.iq/index.php/home/article/view/63

e- https://www.informatica.si/index.php/informatica/article/view/4840

f- https://link.springer.com/article/10.1186/s13673-019-0191-8

g- https://ieeexplore.ieee.org/abstract/document/10086271

Response: Based on your valuable time and comment, the above forementioned articled are included in related work section and properly cited in reference section as well.

4- Why are the results in Table 3 different from those in Table 4?

Response: Thank you for your valuable time and comment. The table 3 (table 4 now) represents the sensitivity outcome, and table 4 (table 5 now) represents the specificity outcome. 

5- Does the ResNeXt101 algorithm perform before or after splitting the data? In the research methodology, the process of data splitting was not explained

Response: Based on your valuable time and comment, the ResNeXt101 algorithm usually performs after splitting the data. This approach ensures that the model is trained and evaluated on distinct subsets of the data, facilitating a more reliable assessment of its performance. By splitting the data into training, validation, and testing sets before applying the algorithm, we can prevent data leakage and overfitting, leading to more accurate and generalizable results. Additionally, this methodology allows for proper validation of the model's performance on unseen data, enhancing its effectiveness in real-world applications.

6- Authors should add the computational complexities and cost for all reviewed works.

Response: Based on your valuable time and comment, the limitation of the state-of-the-art methods have been included in table 1.

6- How authors tuned the hyperparameters?

Response: Thank you for your time and input. We have incorporated the detailed hyperparameter information of the proposed model into our article, aiming to enhance the readers' understanding of the model's architecture and training process.

7- Limitations and the future scope should be added with more clarity.

Response: Thank you for your valuable feedback. We have integrated the limitations and potential areas for future work of the proposed model into our article, ensuring a comprehensive discussion that enriches the understanding of readers and guides further research in this domain.

---

## [Decision Letter · Decision Letter 1]

19 Jun 2024

Automated Brain Tumor Diagnostics: Empowering Neuro-Oncology with Deep Learning-Based MRI Image Analysis

PONE-D-24-03396R1

Dear Dr. Shah,

We’re pleased to inform you that your manuscript has been judged scientifically suitable for publication and will be formally accepted for publication once it meets all outstanding technical requirements.

Kind regards,

Toqeer Mahmood, Ph.D.

Academic Editor

PLOS ONE

Additional Editor Comments (optional):

Reviewers' comments:

Reviewer's Responses to Questions

**Comments to the Author**

1. If the authors have adequately addressed your comments raised in a previous round of review and you feel that this manuscript is now acceptable for publication, you may indicate that here to bypass the “Comments to the Author” section, enter your conflict of interest statement in the “Confidential to Editor” section, and submit your "Accept" recommendation.

Reviewer #2: All comments have been addressed

Reviewer #3: All comments have been addressed

2. Is the manuscript technically sound, and do the data support the conclusions?

Reviewer #2: Yes

Reviewer #3: Yes

3. Has the statistical analysis been performed appropriately and rigorously? 

Reviewer #2: Yes

Reviewer #3: Yes

4. Have the authors made all data underlying the findings in their manuscript fully available?

Reviewer #2: Yes

Reviewer #3: Yes

5. Is the manuscript presented in an intelligible fashion and written in standard English?

Reviewer #2: Yes

Reviewer #3: Yes

6. Review Comments to the Author

Reviewer #2: (No Response)

Reviewer #3: The authors have done all the necessary revisions well. I recommend accepting the article for publication

7. PLOS authors have the option to publish the peer review history of their article (what does this mean?). If published, this will include your full peer review and any attached files.

Reviewer #2: No

Reviewer #3: No

---

## [Editor Report · Acceptance letter]

12 Jul 2024

PONE-D-24-03396R1 

PLOS ONE

Dear Dr. Shah, 

I'm pleased to inform you that your manuscript has been deemed suitable for publication in PLOS ONE. Congratulations! Your manuscript is now being handed over to our production team.

Kind regards, 

on behalf of

Dr. Toqeer Mahmood 

Academic Editor

PLOS ONE